# RECALL-EXTEND DYNAMICS: ENHANCING SMALL LANGUAGE MODELS THROUGH CONTROLLED EXPLORATION AND REFINED OFFLINE INTEGRATION

## ABSTRACT

Many existing studies have achieved significant improvements in the reasoning capabilities of large language models (LLMs) through reinforcement learning with verifiable rewards (RLVR), while the enhancement of reasoning abilities in small language models (SLMs) has not yet been sufficiently explored. Combining distilled data from larger models with RLVR on small models themselves is a natural approach, but it still faces various challenges and issues. Therefore, we propose Recall-Extend Dynamics(RED): Enhancing Small Language Models through Controlled Exploration and Refined Offline Integration. In this paper, we explore the perspective of varying exploration spaces, balancing offline distillation with online reinforcement learning. Simultaneously, we specifically design and optimize for the insertion problem within offline data. By monitoring the ratio of entropy changes in the model concerning offline and online data, we regulate the weight of offline-SFT, thereby addressing the issues of insufficient exploration space in small models and the redundancy and complexity during the distillation process. Furthermore, to tackle the distribution discrepancies between offline data and the current policy, we design a sample-accuracy-based policy shift mechanism that dynamically chooses between imitating offline distilled data and learning from its own policy. Code is available at: `https://anonymous.4open.science/r/OpenRLHF-Millioniron--1709`

## 1 INTRODUCTION

Recent studies (Jaech et al., 2024; Guo et al., 2025; Team et al., 2025) indicate that LLMs can achieve significant performance improvements in various tasks through enhanced reasoning capabilities. A key technology driving this progress is RLVR (Lambert et al., 2024; Shao et al., 2024), which has garnered widespread attention in the research community and led to the development of various policy optimization algorithms such as GRPO (Guo et al., 2025), DAPO (Yu et al., 2025), GPG (Chu et al., 2025), and DR.GRPO (Liu et al., 2025), and others (Chen et al., 2025a; Zhang et al., 2025a). These methods are characterized by their simplicity, directness, and result-oriented approach, distinguishing them from traditional methods based on process reward modeling (Li & Li, 2025; Setlur et al., 2024).

Despite the promising advancements in the aforementioned reasoning-related research, most studies have focused on models ranging from 7B to 32B in parameters. For smaller models (e.g., 1.5B), research has primarily adopted direct or multi-stage distillation methods (Yang et al., 2025; Guo et al., 2025; Abdin et al., 2025), and their integration with RLVR remains largely unexplored.

However, small models trained solely through Supervised Fine-Tuning (SFT) distillation still exhibit noticeable issues such as overthinking and redundant generation (Zhang et al., 2025d; Li et al., 2025), indicating room for overall performance improvement. As illustrated in the Figure 1, pre-SFT models optimized with post-RL show improved reasoning performance and can, to some extent, reduce the probability of generating overthinking words, thereby decreasing redundant output.

Nevertheless, the "pre-SFT + post-RL" training pipeline still faces efficiency challenges. As depicted in the Figure 1, the model's generation length during RL training gradually decreases from

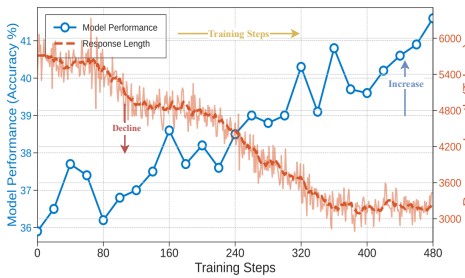
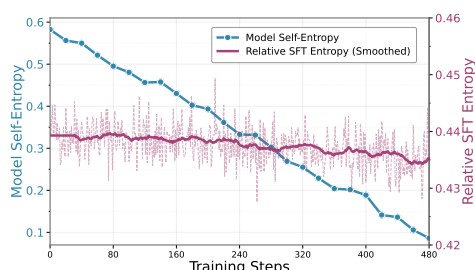

Figure 1: RL training on an SFT-distilled model. The pre-SFT model overthinks , generating a lot of redundant content that leads to inefficient training.

Figure 2: Entropy trends during model training. Blue curve: model self-entropy. Magenta curve: relative SFT trajectory entropy , showing alignment with SFT.

an initial over 6,000 tokens to approximately 3,000, while the rollout time, influenced by generation length, significantly increases, further prolonging the training cycle.

Therefore, this paper proposes a more efficient strategy combining offline-SFT with online-RL, aiming to enhance the reasoning capabilities of small-scale models. Compared to existing SFT + RL methods (Yan et al., 2025; Fu et al., 2025; Zhang et al., 2025c; Huang et al., 2025), our innovation lies in exploring changes in the exploration space to balance offline distillation data with online reinforcement learning, alongside specifically designing and optimizing for the integration problem in offline data.

(I) Controlled Exploration: Balancing Recall and Extend via **Dynamic Entropy Regulation**

Existing research has confirmed that RLVR, in most cases, does not endow models with new capabilities but rather acts as an "amplifier", uncovering and activating existing knowledge from the model's pre-training phase (Yue et al., 2025; Shao et al., 2024). Specifically, RLVR demonstrates performance improvement for smaller $k$ in pass@$k$, but its effect declines for larger $k$. We define this process as a **Recall** phase, where the core objective is to optimize the reasoning path within existing capabilities while contracting the exploration space. SFT, conversely, can expand the reasoning boundary, introducing new reasoning patterns learned from stronger teacher models and increasing the model's explorable space, thus seen as an **Extend** of the model's capabilities (Guo et al., 2025; Kim et al., 2025).

These two approaches can complement each other to some extent. We propose using the entropy variation ratio, an intuitive metric for explorable space, to balance the Recall and Extend phases, addressing the issues of insufficient RL exploration space in small models and the redundancy and complexity problems inherent in the distillation process, thereby achieving effective complementarity between the two.

(II) Adaptive Integration of Offline Data with Accuracy-aware Policy Shifts

Integrating distilled offline data into the policy optimization function can achieve a unified training paradigm. However, using a fixed clip value makes it difficult to handle distillation data that often deviates significantly from the current policy(Appendix F). Furthermore, if distillation samples are treated as a deterministic policy (i.e., setting $\pi_{\text{offline}} = 1$), it can rapidly exacerbate entropy collapse; conversely, directly integrating distillation samples via SFT loss or an on-policy form may lead to model performance collapse.

To address these issues, we propose a method that dynamically adjusts the policy offset based on the answer's correctness rate. For samples with high correctness rates, we prefer setting $\pi_{\text{offline}} = 1$, allowing the model to learn from its own policy; for samples with low correctness rates, the policy offset is set closer to $\pi$, encouraging the model more inclined to imitate the distillation samples. This method not only improves the model's adaptability and robustness to data of varying quality but also optimizes overall training efficiency. In brief, our contributions can be summarized as follows:

• We frame the integration of RL and SFT as a synergy process of Recall and Extend. By dynamically balancing these two stages, we enable Extend phase to satisfy the exploration space required

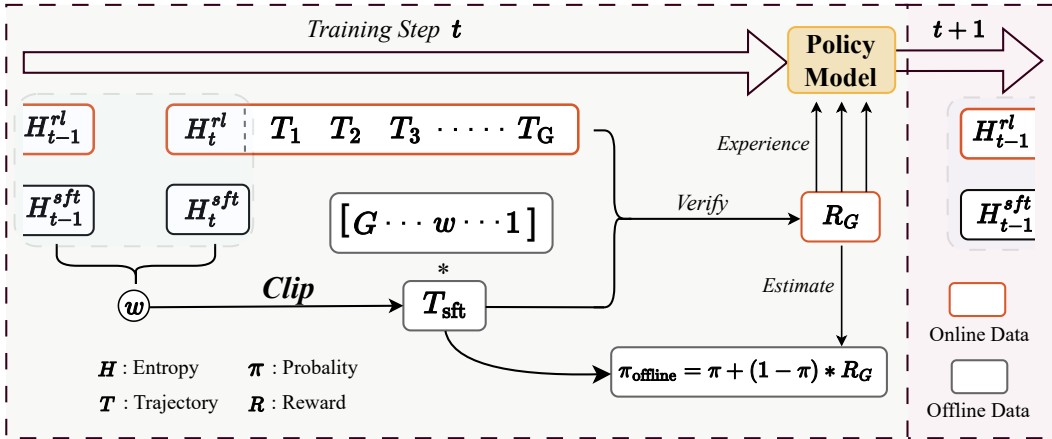

Figure 3: The training pipeline of RED. RED computes the weight $w$ for SFT based on the entropy dynamics of RL and SFT, thereby balancing the contributions of SFT and RL. Additionally, the off-policy probability $\pi_{\text{offline}}$ is determined according to the sample accuracy — the lower the accuracy, the higher the uncertainty, and vice versa.

by Recall phase, while Recall refines and simplifies the complex information from Extend. This achieves steady improvement in smaller models using offline distilled data.

- Our approach significantly improves integration of offline distilled data into the RL process. We introduced a policy offset term based on Accuracy-aware to estimate $\pi_{\text{offline}}$. This mechanism allows the model to adjust autonomously when processing samples with high accuracy, while leaning more towards imitating the large model's strategy for samples with high error rates. This effectively avoids entropy collapse and performance degradation.

## 2 RELATED WORK

### 2.1 REASONING WITH INTEGRATED SFT AND RL

UFT (Wang et al., 2024) proposes a new method that combines the training paradigms of DPO (Rafailov et al., 2023) and SFT, directly aligning the implicit rewards under the DPO's Sigmoid functio with true labels. NFT (Chen et al., 2025b) demonstrates that by additionally utilizing negative samples, the performance gap between SFT and mainstream RL algorithms can be significantly narrowed. In ReLIFT (Ma et al., 2025), researchers collected difficult problems during the RL phase and conducted targeted fine-tuning during the SFT phase, effectively combining RL and SFT to enhance LLM's reasoning and out-of-distribution generalization capabilities, achieving performance beyond the model's inherent cognitive constraints. SASR (Chen et al., 2025c) proposes an adaptive hybrid training framework that dynamically selects SFT or RL training strategies based on the gradients of SFT and GRPO tasks, achieving a dynamic balance between the two throughout the optimization process. BREAD (Zhang et al., 2025c) points out that the inherent performance of small models affects the synergistic effect of SFT and RL. This method introduces branch roll-out to coordinate the two-stage training; for completely failed problems, concise expert prefixes or prompts will be adaptively inserted in the next stage to guide the small model to complete the remaining reasoning path. At a more fundamental level, LUFFY (Yan et al., 2025) introduces distillation data from strong reasoning models, modeling it as off-policy guidance, unified within a zero-RL training paradigm, and integrating the advantages of SFT and RL in a mix-policy form. Furthermore, SRFT (Fu et al., 2025), to more effectively utilize SFT for example learning and RL for policy exploration, proposes an entropy-aware weighting mechanism, organically combining the two fine-tuning paradigms to construct a unified training framework.

## 2.2 SMALL MODEL REASONING

Early community efforts primarily adopted direct distillation to endow small models with reasoning capabilities (Yang et al., 2025; Guo et al., 2025). Furthermore, (Xu et al., 2025a)proposed a systematic training scheme for SLMs, improving small model performance through a hybrid of SFT and reinforcement learning (RL) stages. ADPA(Gao et al., 2025) optimizes the student model's policy by estimating the advantage function between a trained preference-aligned teacher model and a reference model, providing distribution-level reward signals. REDI (Xu et al., 2025b) introduces negative example traces to enhance reasoning distillation, proposing an asymmetrically weighted reference-free objective, which brings performance gains without expensive online reinforcement learning interactions. Chen et al. (2025e) further proposed a re-distillation strategy, generating new SFT training data by sampling from a converged policy to improve model capability. Kim et al. (2025)'s research found that while distillation can improve accuracy, its true potential for raising the model's upper capacity limit depends on introducing new knowledge. If only existing reasoning patterns are distilled, similar to RLVR, it may instead reduce accuracy for some problems. Zhang et al. (2025d) found in extensive experiments that SFT-based distillation methods lead to severe inefficiency in small models. This inefficiency primarily stems from redundant generation and repetitive content, especially under increased computational budgets during testing, resulting in non-monotonic improvements in accuracy.

## 3 PRELIMINARY

In standard GRPO, for each query $q \sim \mathcal{D}$, the model samples a set of responses $O = \{o_1, o_2, \cdots, o_G\}$ from the old policy $\pi_{\theta_{\text{old}}}$, and optimizes the policy model $\pi_\theta$ by maximizing the following objective:

$$\mathcal{J}(\theta) = \mathbb{E}_{(q,a)\sim\mathcal{D}, \{o_i\}_{i=1}^G \sim \pi_\theta(\cdot|q)} \left[ \frac{1}{G} \sum_{i=1}^G \frac{1}{|o_i|} \sum_{t=1}^{|o_i|} \right.$$

$$\left. \left\{ \min\left[ r_{i,t}(\theta)\hat{A}_{i,t}, \text{clip}\left(r_{i,t}(\theta), 1-\epsilon, 1+\epsilon\right)\hat{A}_{i,t} \right] - \beta \, \mathbb{D}_{\text{KL}}\left[\pi_\theta \,\|\, \pi_{\text{ref}}\right] \right\} \right]. \tag{1}$$

Here, $r_{i,t}(\theta)$ is the policy ratio at token $t$ of response $o_i$, $\hat{A}_{i,t}$ is the advantage estimate, and $\epsilon$ controls the clipping range.

Let us further simplify the expression by removing the KL divergence term and assuming that $r_{i,t}(\theta)$ is always within the clipping range.

$$\mathcal{J}(\theta) = \mathbb{E}_{(q,a)\sim\mathcal{D}, \{o_i\}_{i=1}^G \sim \pi_\theta(\cdot|q)} \left[ \frac{1}{G} \sum_{i=1}^G \frac{1}{|o_i|} \sum_{t=1}^{|o_i|} r_{i,t}(\theta)\hat{A}_{i,t} \right]. \tag{2}$$

Taking the gradient of the objective function $\mathcal{J}(\theta)$ yields:

$$\nabla\mathcal{J}(\theta) = \mathbb{E}_{(q,a)\sim\mathcal{D}, \{o_i\}_{i=1}^G \sim \pi_\theta(\cdot|q)} \left\{ \frac{1}{G} \sum_{i=1}^G \frac{1}{|o_i|} \sum_{t=1}^{|o_i|} r_{i,t}(\theta)\hat{A}_{i,t} \nabla\log\pi_\theta(o_{i,t}|q, o_{i,<t}) \right\}. \tag{3}$$

When performing SFT on a small model using distilled reasoning data $(q, o_{\text{sft}}) \sim \mathcal{D}_{\text{sft}}$, the loss function takes the form of a maximum likelihood objective. Supervised learning fundamentally aims to train a model $\pi_\theta(o \mid q)$ to approximate the underlying data distribution $\pi_{sft}(o^{sft} \mid q)$. This can be achieved by optimizing the following maximum likelihood objective:

$$\nabla\mathcal{L}_{\text{sft}}(\theta) = \mathbb{E}_{(q,o^{sft})\sim\mathcal{D}_{\text{sft}}} \left\{ \frac{1}{|o^{sft}|} \sum_{t=1}^{|o|} \nabla\log\pi_\theta(o_t \mid q, o_{<t}) \right\}. \tag{4}$$

In the post-training stage, a simple loss function form of the training framework that unifies RL and SFT can be written as:

$$\nabla \mathcal{J}_{\text{mix}}(\theta) = \mathbb{E}_{(q,o^{sft}) \sim \mathcal{D}, \{o_i\}_{i=1}^G \sim \pi_\theta(.|q)}$$

$$\left\{ \frac{1}{G} \sum_{i=1}^{G} \frac{1}{|o_i|} \sum_{t=1}^{|o_i|} r_{i,t}(\theta) \hat{A}_{i,t} \nabla \log \pi_\theta(o_t \mid q, o_{<t}) + \frac{1}{|o^{sft}|} \sum_{t=1}^{|o^{sft}|} \nabla \log \pi_\theta(o_t^{sft} \mid q, o_{<t}) \right\}. \tag{5}$$

## 4 METHOD

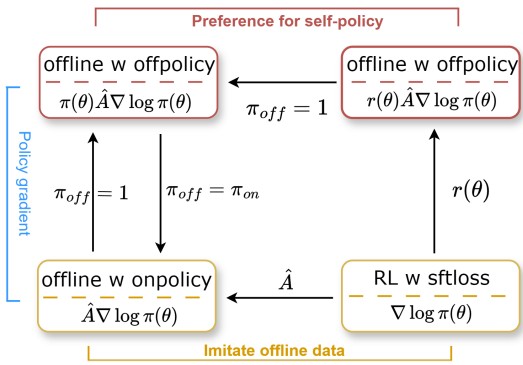

Figure 4: Transformations Between Different Forms of Offline-SFT Integrated into the RL Policy Function.

In this section, we provide a detailed description of our approach, which is divided into two main parts: (I)Balancing Recall and Extend via Dynamic Entropy Regulation, and (II)Adaptive Integration of Offline Data with Accuracy-aware Policy Shifts.

### 4.1 ADAPTIVE INTEGRATION OF DISTILLED DATA

To effectively leverage the advantages of offline-SFT and online-RL, it is essential to integrate their respective objective (loss) functions. In the Preliminary, Eq.(5) presents a simple fusion of RL and SFT. Building on this, an intuitive solution is to incorporate the generated trajectory of the distilled model into the policy optimization function. Furthermore, we illustrate the relationship between these strategies in Figure 4. By adding an advantage term, RL with SFT loss can be easily transformed into on-policy format,

$$\nabla \mathcal{J}_{\text{mix}}(\theta) = \mathbb{E}_{(q,o^{sft}) \sim \mathcal{D}, \{o_i\}_{i=1}^G \sim \pi_\theta(.|q)} \left\{ \frac{1}{G+1} \sum_{i=1}^{G} \frac{1}{|o_i|} \sum_{t=1}^{|o_i|} r_{i,t}(\theta) \hat{A}_{i,t} \nabla \log \pi_\theta(o_t \mid q, o_{<t}) \right.$$
$$\left. + \frac{1}{G+1} \frac{1}{|o^{sft}|} \sum_{t=1}^{|o^{sft}|} \hat{A}_{i,t} \nabla \log \pi_\theta(o_t^{sft} \mid q, o_{<t}) \right\}. \tag{6}$$

And further incorporating importance sampling results in an off-policy formulation.

$$\nabla \mathcal{J}_{\text{mix}}(\theta) = \mathbb{E}_{(q,o^{sft}) \sim \mathcal{D}, \{o_i\}_{i=1}^G \sim \pi_\theta(.|q)} \left\{ \frac{1}{G+1} \sum_{i=1}^{G} \frac{1}{|o_i|} \sum_{t=1}^{|o_i|} r_{i,t}(\theta) \hat{A}_{i,t} \nabla \log \pi_\theta(o_t \mid q, o_{<t}) \right.$$
$$\left. + \frac{1}{G+1} \frac{1}{|o^{sft}|} \sum_{t=1}^{|o^{sft}|} r_{i,t}^{\text{offline}} \hat{A}_{i,t} \nabla \log \pi_\theta(o_t^{sft} \mid q, o_{<t}) \right\}, \tag{7}$$

where $r_{i,t}^{\text{offline}} = \frac{\pi}{\pi^{\text{offline}}}$.

Here, $\pi^{\text{offline}}$ denotes the probability assigned by the larger model's probability for the current trajectory. However, obtaining $\pi^{\text{offline}}$ from a large model is costly, especially with large-scale distillation data. Furthermore, due to inconsistencies in model architectures and differences in vocabularies between large and small models, acquiring an accurate $\pi^{\text{offline}}$ is challenging.

Conversely, using a naive assignment for $\pi^{\text{offline}}$ leads to various issues(Appendix H). When $\pi^{\text{offline}} = 1$, the model's learning from offline data effectively defaults to relying excessively on its own policy($r_{i,t}^{\text{offline}} = \pi$), This causes both SFT entropy and RL entropy to collapse rapidly. When $\pi^{\text{offline}}$ is switched to $\pi$, the model treats offline trajectories as on-policy, this offline-onpolicy approach causes model performance to collapse in the mid-training phase. This demonstrates that the value of $\pi^{\text{offline}}$ is crucial for the integration of offline-SFT.

To address this issue, we propose an **accuracy-aware policy shift mechanism** that dynamically estimates $\pi^{\text{offline}}$ based on sample difficulty. Our empirical analysis (Figure 5) reveals two key phenomena: (1) the model's policy probability $\pi$ correlates positively with sample accuracy — higher

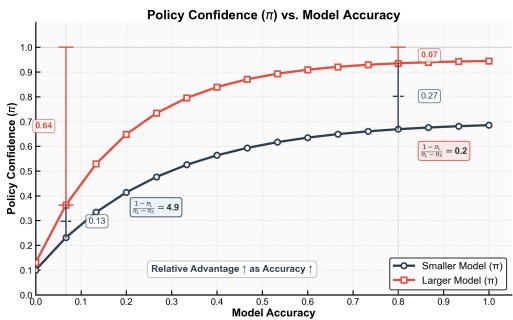 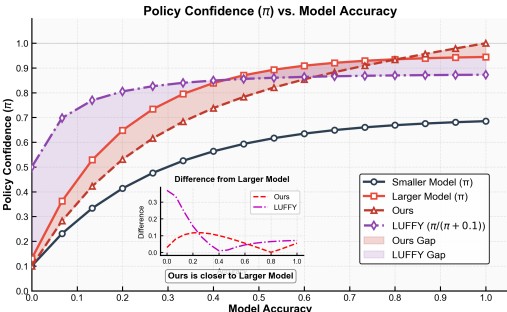

Figure 5: As accuracy rises, policy confidence grows, yet the gap between large and small models widens.

Figure 6: Our calibrated policy ($\pi^{\text{offline}}$) better aligns with the large model's behavior than LUFFY.

accuracy samples yield higher $\pi$ values; and (2) on extremely difficult samples, the policy probabilities of large (Qwen2.5-32B) and small (Qwen2.5-1.5B) models converge, whereas on high-accuracy samples, the gap between them widens significantly, with the large model's $\pi$ approaching 1.

This observation aligns with our intuition: higher problem difficulty implies greater uncertainty, which should correspond to a smaller $\pi$. To formalize this, we introduce the following adaptive estimation:

$$\pi^{\text{offline}} = \pi + (1 - \pi) \cdot r_{\text{group}} = r_{\text{group}} + (1 - r_{\text{group}}) \cdot \pi. \tag{8}$$

For samples with high accuracy, we tend to set $\pi^{\text{offline}} = 1$, allowing the model to adapt based on its own circumstances. Conversely, for samples with high error rates, $\pi^{\text{offline}}$ tends toward $\pi$, making the model more inclined to imitate the large model's policy. As illustrated in Figure 6, our estimated $\pi^{\text{offline}}$ closely tracks the true policy distribution of the large model. Crucially, our method demonstrates superior robustness on samples that are typically under-learned by standard GRPO-style algorithms — namely, extremely easy or extremely hard instances — thereby enhancing training stability and final performance.

We establish three formal propositions that characterize the stability, correctness, and convergence of our offline policy estimator under adaptive importance weighting.

**Proposition 1 (Stability and Adaptivity).** Given the offline policy estimator $\pi_{\text{offline}}(a_t|s_t) = \beta + (1 - \beta)\pi_\theta(a_t|s_t)$ and the corresponding importance weight $\rho_t = \frac{\pi_\theta(a_t|s_t)}{\pi_{\text{offline}}(a_t|s_t)}, \beta = r_{\text{group}}$, the following hold for all $s_t, a_t$:

1. **(Bounded Weight)** $\rho_t \leq \frac{1}{1-\beta}$,

2. **(Robustness)** $\lim_{\pi_\theta(a_t|s_t) \to 0} \rho_t = 0$,

3. **(Entropy Preservation)** For $\beta \to 1$, $\Delta H(\pi_\theta) > 0$ under gradient update.

These properties ensure training stability, robustness to noisy tokens, and prevention of entropy collapse. Proof in Appendix B.1.

**Proposition 2 (Gradient Equivalence).** For hard samples ($\beta \to 0$), the expected policy gradient under our estimator equals that of behavioral cloning under the true teacher policy $\pi^*$:

$$\mathbb{E}_{a_t \sim \pi^*}\left[\rho_t \nabla_\theta \log \pi_\theta(a_t|s_t)\right] = \mathbb{E}_{a_t \sim \pi^*}\left[\nabla_\theta \log \pi_\theta(a_t|s_t)\right]. \tag{9}$$

This confirms that our estimator provides correct learning signals for difficult samples. Proof in Appendix B.2.

**Proposition 3 (Asymptotic Consistency).** Assume: (i) sufficient state-action coverage under $\pi^*$, (ii) $\beta_k \to 0$ as $k \to \infty$, and (iii) learning rates $\{\alpha_k\}$ satisfy $\sum_k \alpha_k = \infty, \sum_k \alpha_k^2 < \infty$. Then:

$$\pi_{\theta_k} \xrightarrow{\text{a.s.}} \pi^* \quad \text{as } k \to \infty. \tag{10}$$

This guarantees convergence to the optimal policy under standard stochastic approximation conditions. Proof in Appendix B.3. Given Propositions 1–3, we optimize the student policy via the following adaptive importance-weighted objective:

$$\mathcal{L}(\theta) = \mathbb{E}_{(s_t, a_t) \sim \mathcal{D}} \left[ \rho_t \cdot \log \pi_\theta(a_t | s_t) \right], \tag{11}$$

where $\rho_t = \frac{\pi_\theta(a_t | s_t)}{\beta + (1-\beta)\pi_\theta(a_t | s_t)}$ is treated as a fixed, precomputed weight during optimization (not a parameter). Due to Propositions 1 and 2, Eq. 11 provides bounded gradients and asymptotically unbiased updates toward $\pi^*$.

## 4.2 CONTROLLED EXPLORATION: BALANCING RECALL AND EXTEND VIA DYNAMIC ENTROPY REGULATION

Our direct motivation stems from the observation that when the exploration space of RLVR is insufficient or incapable of addressing current problems, SFT should be introduced to expand the effective exploration space—and this expansion should occur gradually. During SFT+RL training, RLVR serves as a Recall mechanism, refining reasoning paths within the model's current capabilities while reducing the exploration space. In contrast, SFT acts as an Extend mechanism, enabling the model to acquire out-of-domain knowledge and increase the range of exploration.

Table 1: Entropy evolution across training phases and model checkpoints

| Training | Initial | Step 0 | | Step 300 | |
|---|---|---|---|---|---|
| | | RL | SFT | RL | SFT |
| **RL Phase** | Base | 1.20 | 1.05 | 0.22 | 0.99 |
| | SFT-50 | 0.88 | 0.67 | 0.35 | 0.65 |
| | SFT-100 | 0.56 | 0.44 | 0.30 | 0.43 |
| | SFT-300 | 0.47 | 0.34 | 0.32 | 0.34 |
| **SFT Phase** | Base | 1.20 | 1.05 | 0.47 | 0.34 |
| | RL-50 | 0.44 | 0.90 | 0.44 | 0.33 |
| | RL-100 | 0.36 | 0.89 | 0.45 | 0.30 |
| | RL-300 | 0.30 | 0.88 | 0.44 | 0.30 |

By monitoring the changes in the exploration space( Entropy :$H = -\sum_{i=1}^{|V|} p_i \log p_i$), we dynamically adjust the contributions of offline-SFT and RLVR, aiming to balance the reduction in exploration from Recall and the expansion from Extend. Unlike prior work that directly uses entropy values as weighting signals, we instead use the change in entropy from RL stages as an indicator of exploration dynamics($\Delta H^{rl} = |1 - \frac{H_t^{rl}}{H_{t-1}^{rl}}|, \Delta H^{sft} = |1 - \frac{H_t^{sft}}{H_{t-1}^{sft}}|$).

When the change in RL entropy is small, we increase the contribution of offline-SFT to boost the exploration space and improve the model's performance ceiling. Conversely, when RL entropy exhibits sufficient change (indicating active exploration), the influence of offline-SFT is reduced. We further incorporate changes in SFT entropy to balance the emphasis between RL and SFT. Specifically, $w = \frac{\Delta H^{sft}}{\Delta H^{rl}}$. Crucially, by leveraging the relative dynamics of entropy changes—rather than static entropy values—our method enables adaptive weighting of SFT and RL throughout the entire training process (early, middle, and late stages). This avoids the common pitfall in prior approaches of blindly reinforcing SFT in later stages regardless of the actual exploration state appendix H.

It is noted that, as shown in the Table 1, as training progresses, RL training leads to a decrease in RL entropy but does not cause a change in SFT entropy. In contrast, normal SFT causes changes in both SFT entropy and RL entropy. Therefore, by solely adjusting the weight for offline-SFT, we can achieve fine-grained control over the exploration space.

$$\nabla \mathcal{J}_{\text{mix}}(\theta) = \mathbb{E}_{(q, o^{sft}) \sim \mathcal{D}, \{o_i\}_{i=1}^G \sim \pi_\theta(\cdot|q)} \left\{ \frac{1}{G+1} \sum_{i=1}^G \frac{1}{|o_i|} \sum_{t=1}^{|o_i|} r_{i,t}(\theta) \hat{A}_{i,t} \nabla \log \pi_\theta(o_t | q, o_{<t}) \right.$$

$$\left. + \frac{w}{G+1} \frac{1}{|o^{sft}|} \sum_{t=1}^{|o^{sft}|} r_{i,t}^{\text{offline}} \hat{A}_{i,t} \nabla \log \pi_\theta(o_t^{sft} | q, o_{<t}) \right\}. \tag{12}$$

To ensure numerical stability and interpretability, we apply an upper and lower clip to the computed weight within the range $[1, G]$, where $G$ is the number of samples in a group. When the weight is 1, it implies that the offline-SFT update acts on a single sample within the group; when it equals $G$, it

| Model | AIME24 | | AIME25 | | AMC | | MATH500 | | Minerva | | Olympiad | | Overall | |
|---|---|---|---|---|---|---|---|---|---|---|---|---|---|---|
| | *Acc* | *LEN* | *Acc* | *LEN* | *Acc* | *LEN* | *Acc* | *LEN* | *Acc* | *LEN* | *Acc* | *LEN* | *Acc* | *LEN* |
| SFT | 13.54 | 7254 | 16.78 | 7011 | 50.23 | 5170 | 72.6 | 3354 | 26.8 | 4742 | 35.7 | 5503 | 35.94 | 5505 |
| GRPO | 9.9 | 2327 | 8.9 | 2344 | 45.7 | 1540 | 71.5 | 1300 | 28.7 | 1205 | 27.1 | 1840 | 31.96 | 1759 |
| SFT+DAPO | 16.6 | 4474 | **18.2** | 3878 | **58.4** | 2783 | 78.6 | 2124 | 33.8 | 3684 | 37.9 | 3866 | 40.58 | 3468 |
| MixPolicy | 14.05 | 2357 | 10.73 | 2001 | 53.04 | 1606 | 75.4 | 1074 | 33.5 | 1129 | 35.7 | 1711 | 37.06 | 1646 |
| Unified | | | | | | | | | | | | | | |
| LUFFY | 15.4 | 3717 | 13.3 | 3017 | 57.34 | 2408 | 78.3 | 1982 | 33.5 | 2216 | 40.6 | 2614 | 39.74 | 2659 |
| SRFT* | 13.7 | 3344 | 12.1 | 2575 | 53.4 | 2132 | 75.8 | 1650 | 34.3 | 2113 | 36.9 | 2557 | 37.7 | 2395 |
| Stage-wise | | | | | | | | | | | | | | |
| ReLIFT | 13.1 | 2236 | 8.9 | 1912 | 50.0 | 1416 | 73.6 | 1086 | 24.6 | 1702 | 34.7 | 1605 | 34.15 | 1659 |
| BREAD* | 13.3 | 2675 | 10.8 | 2034 | 53.6 | 1795 | 74.4 | 1338 | 30.5 | 1752 | 36.1 | 2467 | 36.45 | 2010 |
| Red with (I) | 14.97 | 2288 | 12.45 | 1976 | 52.88 | 1612 | 76.2 | 1053 | 35.2 | 1070 | 36.7 | 1727 | 38.06 | 1621 |
| Red with (II) | 15.73 | 2830 | 13.81 | 2762 | 55.32 | 2007 | 78.3 | 1808 | 30.8 | 1338 | 37.9 | 2309 | 38.64 | 2175 |
| Red(ALL) | **16.8** | 3664 | 16.2 | 2896 | 57.5 | 2047 | **80.8** | 1773 | **37.6** | 2226 | **40.7** | 2433 | **41.6** | 2506 |

Table 2: Overall performance on five competition-level mathematical reasoning benchmarks based on Qwen2.5-Math-1.5B. Bold and underline represent the 1st and 2nd in performance. Models marked with an asterisk (*) do not have publicly available code; their results were reproduced based on their respective papers.

indicates a full parallel fusion of SFT and RL.

$$w = \text{clip}(\frac{\Delta H^{sft}}{\Delta H^{rl}}, 1, G). \tag{13}$$

By employing dynamic entropy regulation, we effectively combine RL to refine reasoning with SFT to broaden the exploration space.

## 5 EXPERIMENTS

We evaluate RED on five challenging mathematical reasoning benchmarks: MATH500, AIME24/25, AMC (Li et al., 2024), Olympiad (He et al., 2024), and Minerva (Lewkowycz et al., 2022). Our base model is Qwen2.5-1.5B-MATH. We compare RED against a comprehensive suite of baselines, including pure SFT, pure GRPO, and hybrid methods such as LUFFY (Yan et al., 2025), ReLIFT (Ma et al., 2025), SRFT (Fu et al., 2025), and BREAD (Zhang et al., 2025c). For evaluation, we use pass@1 for larger test sets and avg@32 for smaller ones (AIME/AMC) to ensure statistical reliability. Full implementation details, including hyperparameters and dataset specifics, are provided in Appendix C.

### 5.1 EXPERIMENTAL RESULTS

**Main Results.** Our results (Table 2) yield four key insights: (1) Under small-model constraints, SFT outperforms pure RL but remains inefficient in token usage. (2) SFT+RL typically complements each other, balancing effectiveness and efficiency. (3) Unified training paradigms (LUFFY, SRFT, RED) surpass stage-separated approaches (ReLIFT, BREAD), validating our design's foundational premise. (4) **RED**, significantly outperforms all baselines, demonstrating clear efficacy.

**Ablation Results.** We conduct an ablation study on RED's key components (Table 2), focusing on two entropy-aware mechanisms: (I) *Dynamic Entropy Regulation*, which balances recall and extension during generation; (II) *Accuracy-aware Offline Dynamic Policy Shifts*, which adapt policy weights based on correctness signals. Results show both components are crucial: Dynamic Entropy Regulation alone yields limited gains, but becomes highly effective when combined with Policy Shifts, confirming that unified SFT/RL is essential for balancing exploration and exploitation.

**Reasoning Efficiency.** Beyond accuracy, we measure average response length (LEN). Surprisingly, two-stage models (ReLIFT, BREAD) generate shorter responses than unified LUFFY/SRFT.We hypothesize that this phenomenon arises because the targeted design of

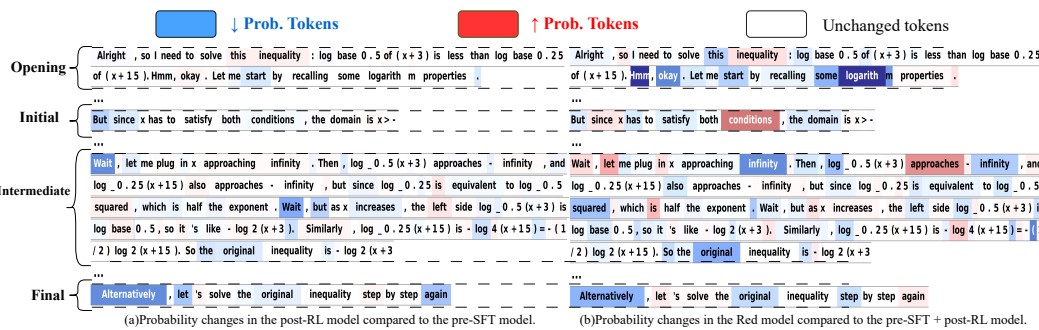

Figure 7: Blue indicates a decrease in probability, red indicates an increase, and white indicates no change, with darker colors signifying a greater degree of change. The reasoning process can be roughly divided into four stages. **Open**: Typically consists of the model's verbalized preamble or introductory statements. **Initial**: Located at the beginning of the reasoning process. **Intermediate**: Constitutes the main body or middle part of the reasoning. **Final**: Serves as the conclusion

the two-stage training approach reduces its susceptibility to offline-sft. RED, however, combines the best of both paradigms, achieving superior efficiency and performance.

Table 3: Evaluation results on out-of-domain datasets. All metrics are accuracy (%).

| Method | ARC-C | GPQA-D | MMLU-Pro | Avg |
|---|---|---|---|---|
| LUFFY | 58.2 | 15.4 | 23.6 | 32.4 |
| SRFT | 56.8 | 16.7 | 16.5 | 30.0 |
| ReLIFT | 55.1 | 10.6 | 16.6 | 27.4 |
| Red(Ours) | 60.4 | 19.8 | 25.4 | 35.2 |

### 5.2 CASE STUDY

To better understand the differences between the RED model and baseline RL/SFT models, we visualized the token probability shifts during reasoning, as shown in Figure 7.

We compare (a) post-RL vs. pre-SFT, and (b) RED vs. post-RL. In Figure 7(a), RL training reduces the likelihood of thinking-related tokens ("But", "Wait", "Alternatively") across all reasoning stages: open, initial, intermediate, and final. While this may shorten reasoning length, it appears to hinder thought initiation, particularly in the initial and intermediate phases. In contrast, Figure 7(b) shows that the RED model increases thinking-related probabilities during initial and intermediate stages, but decreases them in the final stage. This pattern reflects a more efficient reasoning trajectory: deeper exploration early on leads to better-formed conclusions, reducing redundant deliberation at the end. This demonstrates the RED promotes thoughtful analysis when most needed, and decisive closure when appropriate, enhancing reasoning quality and efficiency.

### 5.3 OUT-OF-DOMAIN DATASETS

To further validate the generalization ability of our model, we conduct experiments on three out-of-domain datasets. **ARC-C**: Open-domain scientific reasoning; **GPQA-diamond**: Graduate-level science knowledge ; **MMLU-Pro**: General academic knowledge. Table 3 presents the OOD evaluation results. RED consistently outperforms baselines across all datasets, showing strong out-of-domain generalization beyond MATH.

We also provide ablation studies on hyperparameters (Appendix D), revealing how each component contributes, alongside monitoring metrics tracking its training dynamics (Appendix E), Bias between offline data and SLM(Appendix F), robusness(Appendix G), concurrent work( I).

## 6 CONCLUSION

We introduce _R_ecall-_E_xtend _D_ynamics, a novel method designed to significantly enhance the reasoning capabilities of small language models. It cleverly blends SFT and RL by adaptively balancing when to "extend" knowledge and when to "recall" and refine it based on model "exploration". RED also intelligently integrates distilled data from larger models, allowing SLMs to learn robustly and efficiently.

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

## APPENDIX CONTENTS

# A USE OF LLMS

We utilize LLMs to assist with formula derivations and writing refinement on this paper.

# B DETAILED PROOFS

This appendix provides rigorous proofs for the theoretical propositions presented in the main text.

PROPOSITION A.1 (STABILITY AND ADAPTIVITY)

**Proposition B.1** (Stability and Adaptivity)**.** *Given the offline policy estimator* $\pi_{offline}(a_t|s_t) = \beta + (1 - \beta)\pi_\theta(a_t|s_t)$ *and the corresponding importance weight* $\rho_t = \frac{\pi_\theta(a_t|s_t)}{\pi_{offline}(a_t|s_t)}$*, the following properties hold for all states* $s_t$ *and actions* $a_t$*:*

1. ***Training Stability:*** *The importance weight is bounded:* $\rho_t \leq \frac{1}{1-\beta}$*.*

2. ***Robustness to Low-Confidence Tokens:*** *As* $\pi_\theta(a_t|s_t) \to 0$*, the importance weight* $\rho_t \to 0$*.*

3. ***Entropy Preservation:*** *For easy samples* ($\beta \to 1$)*, the gradient update does not monotonically decrease the student policy's entropy,* $H(\pi_\theta)$*.*

*Proof.* We prove each property in turn.

1. **Training Stability (Bounded $\rho_t$)**

   The importance weight is defined as:

   $$\rho_t = \frac{\pi_\theta(a_t|s_t)}{\beta + (1-\beta)\pi_\theta(a_t|s_t)}$$

   Since $\pi_\theta(a_t|s_t) \in (0, 1]$ and $\beta \geq 0$, the denominator is trivially lower-bounded:

   $$\beta + (1-\beta)\pi_\theta(a_t|s_t) \geq (1-\beta)\pi_\theta(a_t|s_t)$$

   As all terms are positive, taking the reciprocal reverses the inequality:

   $$\frac{1}{\beta + (1-\beta)\pi_\theta(a_t|s_t)} \leq \frac{1}{(1-\beta)\pi_\theta(a_t|s_t)}$$

   Multiplying by $\pi_\theta(a_t|s_t)$ yields the upper bound:

   $$\rho_t = \frac{\pi_\theta(a_t|s_t)}{\beta + (1-\beta)\pi_\theta(a_t|s_t)} \leq \frac{\pi_\theta(a_t|s_t)}{(1-\beta)\pi_\theta(a_t|s_t)} = \frac{1}{1-\beta}$$

   For any $\beta < 1$, the term $\frac{1}{1-\beta}$ is a finite constant. Thus, $\rho_t$ is uniformly bounded, ensuring that the policy gradient, which is scaled by $\rho_t$, remains stable.

2. **Robustness to Low-Confidence Tokens**

   We consider the limit of $\rho_t$ as the student policy's confidence in an action $a_t$ approaches zero, i.e., $\pi_\theta(a_t|s_t) \to 0$:

   $$\lim_{\pi_\theta \to 0} \rho_t = \lim_{\pi_\theta \to 0} \frac{\pi_\theta(a_t|s_t)}{\beta + (1-\beta)\pi_\theta(a_t|s_t)} = \frac{0}{\beta + (1-\beta) \cdot 0} = 0$$

   A vanishing importance weight $\rho_t$ ensures that the gradient contribution from low-confidence actions is suppressed. This provides robustness against noisy or erroneous tokens in the offline dataset.

3. **Entropy Preservation**

The entropy of the student policy is $H(\pi_\theta) = -\mathbb{E}_{s_t \sim \mathcal{S}}[\sum_{a_t \in \mathcal{V}} \pi_\theta(a_t|s_t) \log \pi_\theta(a_t|s_t)]$. For easy samples ($\beta \to 1$), the objective's gradient is $\nabla_\theta J(\theta) = \mathbb{E}[\nabla_\theta \pi_\theta(a_t|s_t)]$. The first-order change in entropy after a gradient step $\theta \leftarrow \theta + \epsilon \nabla_\theta J(\theta)$ is:

$$\Delta H(\pi_\theta) \approx -\epsilon \mathbb{E}\left[\sum_{a_t} \nabla_\theta \pi_\theta(a_t|s_t) \log \pi_\theta(a_t|s_t) + \sum_{a_t} \pi_\theta(a_t|s_t)\frac{\nabla_\theta \pi_\theta(a_t|s_t)}{\pi_\theta(a_t|s_t)}\right]$$

The second sum simplifies to $\sum_{a_t} \nabla_\theta \pi_\theta(a_t|s_t) = \nabla_\theta \sum_{a_t} \pi_\theta(a_t|s_t) = \nabla_\theta(1) = 0$. This leaves:

$$\Delta H(\pi_\theta) \approx -\epsilon \mathbb{E}\left[\sum_{a_t} \nabla_\theta \pi_\theta(a_t|s_t) \log \pi_\theta(a_t|s_t)\right]$$

For an easy sample, the policy update increases the probability for the target action $a_t$ and decreases it for other actions $a' \neq a_t$. For these other actions with low confidence ($\pi_\theta(a'|s_t) \approx 0$), $\nabla_\theta \pi_\theta(a'|s_t)$ is negative, while $\log \pi_\theta(a'|s_t) \to -\infty$. The product $\nabla_\theta \pi_\theta(a'|s_t) \log \pi_\theta(a'|s_t)$ becomes a large positive value. This positive contribution dominates the sum, leading to $\Delta H(\pi_\theta) > 0$. Thus, for easy samples, the update does not cause a monotonic decrease in entropy.

$\square$

PROPOSITION A.2 (GRADIENT EQUIVALENCE)

**Proposition B.2** (Gradient Equivalence). *For hard samples ($\beta \to 0$), the policy gradient induced by our estimator is equivalent to the true behavioral cloning gradient w.r.t. the teacher policy $\pi^*$:*

$$\mathbb{E}_{a_t \sim \pi^*}[\rho_t \nabla_\theta \log \pi_\theta(a_t|s_t)] = \mathbb{E}_{a_t \sim \pi^*}[\nabla_\theta \log \pi_\theta(a_t|s_t)]$$

*Proof.* From Proposition B.1, we established that for hard samples, $\lim_{\beta \to 0} \rho_t = 1$. We analyze the limit of the left-hand side of the equation. Assuming mild regularity conditions that permit the interchange of the limit and expectation (e.g., via the Dominated Convergence Theorem, since $\rho_t$ is bounded), we have:

$$\lim_{\beta \to 0} \mathbb{E}_{a_t \sim \pi^*}[\rho_t \nabla_\theta \log \pi_\theta(a_t|s_t)] = \mathbb{E}_{a_t \sim \pi^*}\left[\lim_{\beta \to 0} \rho_t \cdot \nabla_\theta \log \pi_\theta(a_t|s_t)\right]$$
$$= \mathbb{E}_{a_t \sim \pi^*}[1 \cdot \nabla_\theta \log \pi_\theta(a_t|s_t)]$$
$$= \mathbb{E}_{a_t \sim \pi^*}[\nabla_\theta \log \pi_\theta(a_t|s_t)]$$

This final expression is the standard MLE gradient for behavioral cloning from the true teacher policy $\pi^*$. $\square$

PROPOSITION A.3 (ASYMPTOTIC CONSISTENCY)

**Proposition B.3** (Asymptotic Consistency). *Assume that (i) the offline data provides sufficient coverage of the state-action space of the teacher policy $\pi^*$, (ii) the difficulty calibration hyperparameter $\beta$ is appropriately annealed such that all samples are eventually treated as hard, and (iii) the learning rate schedule satisfies the Robbins-Monro conditions ($\sum_k \alpha_k = \infty$, $\sum_k \alpha_k^2 < \infty$). Then, the student policy $\pi_\theta$ converges to the teacher policy $\pi^*$ almost surely.*

*Proof.* The proof framework is grounded in stochastic approximation theory. Our training objective is to minimize the Kullback-Leibler (KL) divergence $D_{\mathrm{KL}}(\pi^*||\pi_\theta)$, which is equivalent to maximizing the log-likelihood of the teacher's data under the student policy. The gradient of this objective is $\mathbb{E}_{a_t \sim \pi^*}[\nabla_\theta \log \pi_\theta(a_t|s_t)]$.

From Proposition B.2, we know that for hard samples ($\beta \to 0$), our algorithm's expected update direction is an unbiased estimator of this true gradient. The root of this gradient field, where $\nabla_\theta \mathbb{E}[\log \pi_\theta] = 0$, corresponds to the parameters $\theta^*$ for which $\pi_{\theta^*} = \pi^*$.

The training process is a stochastic gradient descent algorithm. Under the Robbins-Monro theorem, a stochastic approximation procedure converges almost surely to the root of the expected update

vector if the stochastic updates are an unbiased estimate of the true gradient and the learning rate schedule satisfies the stated conditions.

The adaptive nature of our method does not introduce any asymptotic bias. As $\beta$ is annealed towards 0 over time, the updates increasingly resemble the true MLE gradient. Therefore, the algorithm is guaranteed to converge to the parameters $\theta^*$ that minimize the KL-divergence, resulting in $\pi_\theta \rightarrow \pi^*$. $\qquad\square$

## C    DETAILED EXPERIMENTAL SETUP

**Dataset.**  For the training dataset, we used OpenR1-Math-46k-8192 as provided by LUFFY (Yan et al., 2025), with prompts sourced from NuminaMath 1.5 (Li et al., 2024) and detailed demonstrations generated by DeepseekR1 (Guo et al., 2025). All trajectory lengths were below 8192 tokens. For the test dataset, we selected MATH500, AIME24/25, AMC (Li et al., 2024), Olympiad (He et al., 2024), and Minerva (Lewkowycz et al., 2022).

**Evaluation.**  For our own models, due to the relatively small test sets for AIME 24/25 and AMC, we report avg@32 for these datasets, with a temperature of 0.6. For MATH500, Minerva, and Olympiad, we used pass@1 as the evaluation metric.

**Baselines.**  We chose Qwen2.5-1.5B-MATH as our base small model for training. The models we compared against include: (1) **SFT**: Only performs SFT on the dataset. (2) **GRPO**: Only uses simple GRPO on the dataset. (3) **SFT+DAPO**: First distills with SFT, then trains with GRPO. (4) **GRPO-mixpolicy**: Simply merges offline data into the policy optimization function. (5) **LUFFY** (Yan et al., 2025): A shaping mixed-policy GRPO approach. (6) **ReLIFT** (Ma et al., 2025): Interleaves RFT and SFT, with the SFT focusing on the problems that RFT finds hard to solve. (7) **SRFT** (Fu et al., 2025): Conducted based on the entropy of the model's RL stage. (8) **BREAD** (Zhang et al., 2025c): Introduces branch rollout to coordinate the two-stage training.

**Implementation Details.**  For our models, following the LUFFY setup, we adopted a learning rate of 5e-5 for SFT training and trained for 3 epochs. For RL training, we used a learning rate of 2e-6, a temperature of 1.0, and performed 8 rollout iterations. We used the AdamW optimizer with a linear learning rate scheduler.

## D    HYPERPARAMETER EXPERIMENTS FOR KEY COMPONENTS

To verify the effectiveness of critical components in Red, we conduct ablative experiments on two core hyperparameters: the *clipping range of weight* $w$ and the *calculation window of entropy change ratio* $\Delta H$. Detailed settings and results are shown in Table 4.

### D.1    WEIGHT CLIPPING RANGE

The weight $w$ in Red is defined to balance the entropy changes from SFT and RL stages, with the following clipping operation:

$$w = \text{clip}\left(\frac{\Delta H^{\text{SFT}}}{\Delta H^{\text{RL}}}, \text{clip}_{\text{low}}, \text{clip}_{\text{high}}\right) \qquad (14)$$

where $\Delta H^{\text{SFT}}$ and $\Delta H^{\text{RL}}$ denote the entropy changes of the model during SFT and RL training, respectively. We compare three clipping strategies: *Bounded (0, G)*: $\text{clip}_{\text{low}} = 0$, $\text{clip}_{\text{high}} = G$ (fixed upper bound of 8); *Bounded (1, $\infty$)*: $\text{clip}_{\text{low}} = 1$, $\text{clip}_{\text{high}} = +\infty$ (no upper bound, fixed lower bound of 1); *Bounded (1, G)* (our default): $\text{clip}_{\text{low}} = 1$, $\text{clip}_{\text{high}} = G$ (where $G = 8$ is an empirically tuned constant).

### D.2    ENTROPY CHANGE CALCULATION WINDOW

The entropy change ratio $\Delta H$ is computed based on the sliding window of historical entropy values. We test three window sizes (baseline vs. smoothed settings): *Baseline*: Window size = 1 (uses the entropy difference between the current step and the previous 1 step); *Smoothed-5*: Window size =

Table 4: Hyperparameter ablations for key components of Red. Evaluation metric is accuracy (%) on two math datasets (AIME25 and MATH500), with "Overall" as the average of the two.

| Hyperparameter Settings | | Accuracy (%) | | |
|---|---|---|---|---|
| Clipping Range | Calculation Window | AIME25 | MATH500 | Overall |
| Bounded $(1, G)$ | Baseline (1-step) | 16.2 | 80.8 | 48.5 |
| Bounded $(0, G)$ | Baseline (1-step) | 15.1 | 77.9 | 46.5 |
| Bounded $(1, +\infty)$ | Baseline (1-step) | 13.9 | 76.9 | 45.4 |
| Bounded $(1, G)$ | Smoothed-5 | 15.8 | 79.2 | 47.5 |
| Bounded $(1, G)$ | Smoothed-10 | 15.5 | 78.5 | 47.0 |

5 (uses the average entropy of the previous 5 steps); *Smoothed-10*: Window size = 10 (uses the average entropy of the previous 10 steps).

### D.3 ANALYSIS

For the entropy change calculation window, the *Baseline (1-step)* window outperforms smoothed alternatives: *Smoothed-5* and *Smoothed-10* cause 1.0% and 1.5% overall accuracy drops, respectively. Larger smoothing windows blur short-term entropy fluctuations, diluting critical real-time training cues needed for math reasoning.

Overall, Red's hyperparameter choices—*Bounded (1, 8)* for $w$ and 1-step window for $\Delta H$—are empirically optimized. These settings maintain SFT-RL entropy balance and capture real-time dynamics, directly supporting the model's strong math performance and validating its core design.

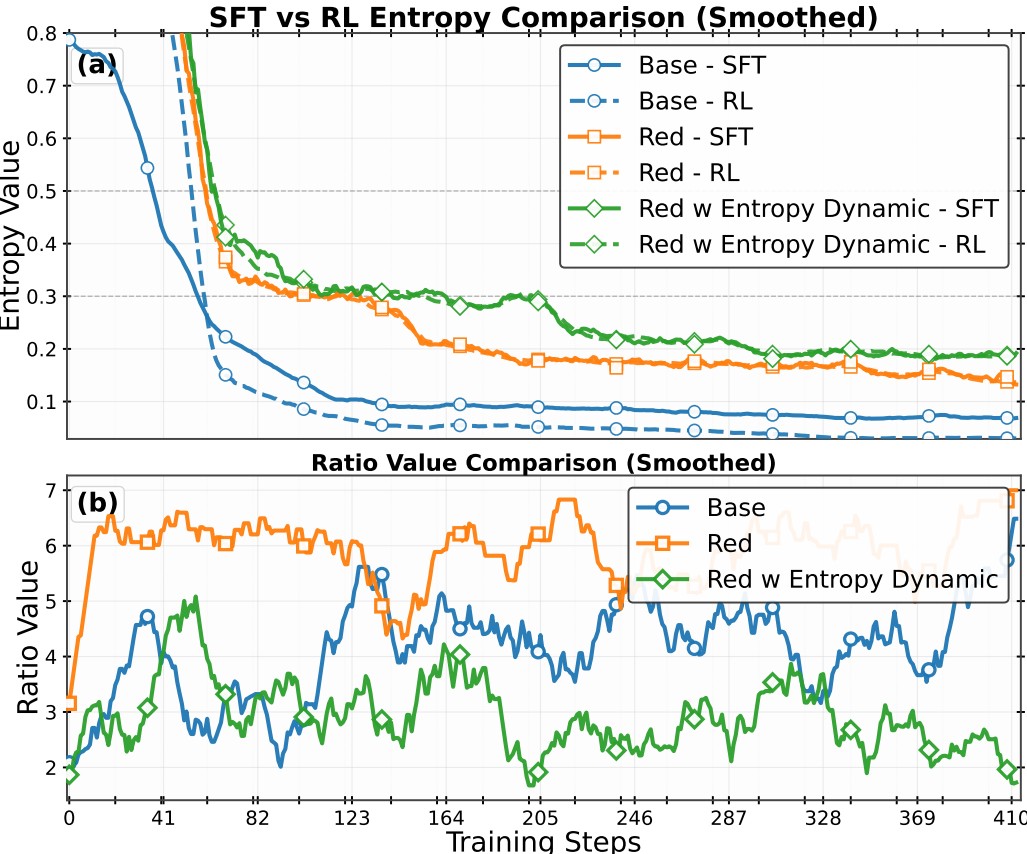

Figure 8: Training dynamics during different training setting , including training sft/rl entropy, ratio
(w = $\Delta H^{sft}/\Delta H^{rl}$).

# E    TRAINING DYNAMICS.

To better illustrate how RED's training progresses, we've visualized the changes in both SFT and
RL entropies as well as their changes in ratio (w = $\Delta H^{rl}/\Delta H^{sft}$) in Figure 6.

Figure 6(a) clearly shows that when we apply Accuracy-aware Policy Shifts, both the RL and SFT
entropies not only increase significantly and remain at a relatively high level, but also are nearly
identical in magnitude. This is a notable improvement compared to simpler approaches like setting
$\pi_{\text{offline}} = 1$, $\pi_{\text{offline}} = \pi$. Higher entropy generally indicates a greater diversity in the model's policy,
allowing for more exploration and less deterministic behavior.

Furthermore, Figure 6(b) highlights the impact of Dynamic Entropy Regulation. After applying
Accuracy-aware Policy Shifts, although the entropy of SFT and RL is effectively adjusted, the fluc-
tuation of their ratio increases and remains relatively high. Through the implementation of Dynamic
Entropy Regulation, the balance between RL and SFT is dynamically and more stably maintained.
This is crucial as it ensures RL retains the necessary exploration space to discover novel strategies
while SFT continues to expand the model's external knowledge, thereby enhancing the robustness
and adaptability of the RED training process.

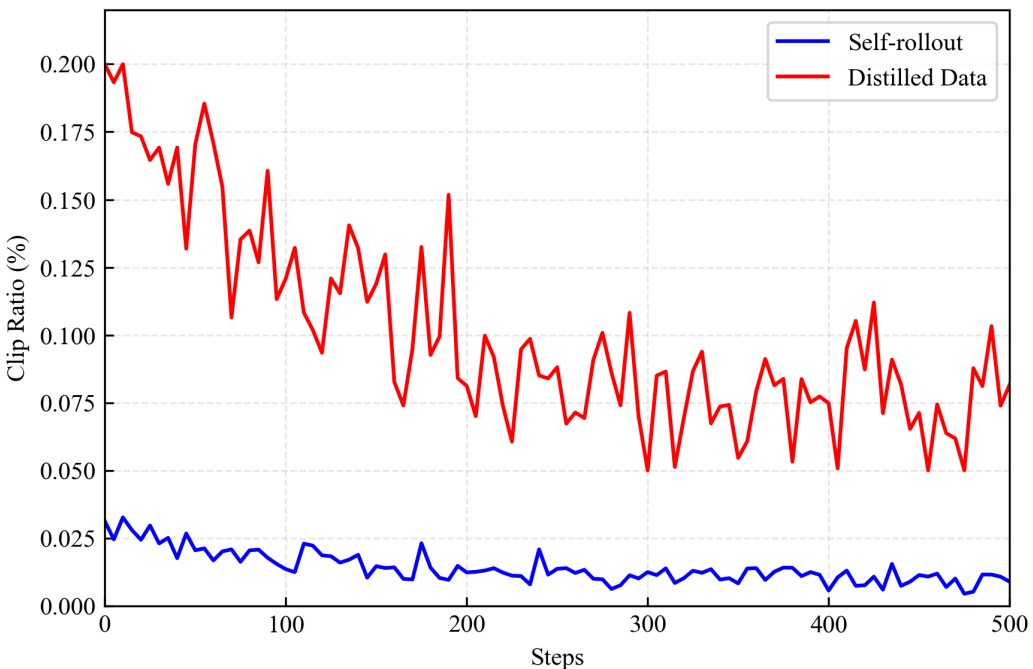

Ratio of clipped signals showing a decreasing trend with slowing rate.

Figure 9: Clip Ratio Comparison: Self-Rollout Online Data vs. Distillation Data

## F  SIGNIFICANT BIAS BETWEEN DISTILLATION DATA AND THE INITIAL POLICY OF SMALL MODELS

This section focuses on quantifying the core discrepancy in bias clip ratio between distillation data and online interaction data, and highlights how this discrepancy poses critical challenges to small-model training.

As illustrated in Figure 9, a pronounced gap emerges in the clip ratio when the small model processes two distinct data sources during iterative learning:

1. Distillation data: Derived from the pre-trained large model's prior knowledge, which encodes idealized decision-making patterns but may not align with real-world environment dynamics;

2. Self-rollout online data: Generated in real time via the small model's direct interaction with the environment, capturing practical feedback (e.g., reward signals, state transitions) but reflecting the model's current imperfections.

This clip ratio mismatch stems from the inherent distribution shif between the two data sources. Left unaddressed, it can lead to severe training instability—including conflicting parameter update directions, delayed convergence, or even divergence when the model attempts to reconcile idealized (distillation) and practical (online) signals. Consequently, mitigating this bias becomes a pivotal priority for optimizing small-model performance.

## G  ROBUSTNESS TO NOISY AND ADVERSARIAL OFFLINE DISTILLATION DATA

In this section, we investigate the robustness of offline distillation when the training data is corrupted by noise or contains adversarial examples that are specifically designed to misalign with the model's reasoning strategy.

We consider two distinct corruption scenarios:

Table 5: Performance under noisy and adversarial offline distillation settings across datasets.

| Dataset | Setting | Clean | Noisy Labels | Adversarial Chains |
|---------|---------|-------|--------------|--------------------|
| MATH500 | Accuracy (%) | 80.8 | 80.6 | 79.5 |
| AIME25 | Accuracy (%) | 16.2 | 16.2 | 15.8 |

1. **Noisy labels**: The distillation dataset contains mislabeled quality scores. Specifically, (i) incorrect model outputs are erroneously assigned high accuracy scores, or (ii) correct outputs with minor annotation discrepancies are incorrectly downgraded to low accuracy scores.

2. **Adversarial reasoning chains**: We construct inputs paired with plausible but logically flawed reasoning trajectories that lead to the correct final answer. These chains appear coherent on the surface yet contain subtle fallacies, thereby challenging the model's ability to discern genuine reasoning from deceptive mimicry.

Our empirical findings are summarized in Table 5. The results reveal two key insights. First, performance degradation under label noise is surprisingly mild—e.g., on MATH500, accuracy drops by only 0.2%—suggesting that the distillation process exhibits a degree of robustness to random or systematic mislabeling of quality scores. This may be attributed to the model's reliance on consensus patterns across the dataset rather than individual score annotations. Second, adversarial reasoning chains induce a more consistent (though still modest) performance drop across both datasets, indicating that models are somewhat susceptible to superficially coherent but logically invalid reasoning. Notably, the absolute impact is larger on the more challenging AIME25 benchmark (0.4% drop vs. 1.3% on MATH500 in relative terms), highlighting that harder problems may be more vulnerable to subtle reasoning perturbations. Overall, these findings suggest that while offline distillation is reasonably robust to data imperfections, adversarial manipulation of the reasoning structure poses a more nuanced threat than mere label noise.

## H    IMPACT OF $\pi^{\text{OFFLINE}}$ ASSIGNMENT

Conversely, a naive choice of $\pi^{\text{offline}}$ leads to significant training instability. When $\pi^{\text{offline}} = 1$, the model effectively treats all offline actions as fully confident, causing the offline reward to collapse to the current policy ($r_{i,t}^{\text{offline}} = \pi(a_t \mid s_t)$). This results in rapid entropy collapse for both SFT and RL stages, as shown in Figure 10.

In contrast, setting $\pi^{\text{offline}} = \pi$, treating offline trajectories as on-policy—introduces severe distributional mismatch during mid-training, leading to performance degradation. These results highlight that the design of $\pi^{\text{offline}}$ is critical for stable integration of offline SFT into the RLHF pipeline.

## I    CONCURRENT WORK

While we previously reviewed representative prior works, our coverage of relevant explorations on the current SFT+RL training paradigm remains incomplete. To address this gap, this section supplements concurrent works published or disclosed in the past month. Notably, including these works here is solely for comprehensive analysis, not for comparing model performance or efficiency—for two key reasons: first, these works are concurrent with ours, their findings not yet fully consolidated, lacking an objective, mature reference basis to justify direct comparisons; second, most lack effective peer review (with methodological rigor and result reliability unvalidated), so comparisons using such outcomes are neither academically rigorous nor practically informative.

CHORD (Zhang et al., 2025b) reframes SFT not as a separate stage but as a dynamically weighted auxiliary objective within the on-policy RL process. To harmonize off-policy expert imitation and on-policy exploration, it introduces a dual-control mechanism: a global coefficient to guide the transition between imitation and exploration, and a token-wise weighting function to enable granular learning from expert tokens while mitigating disruption from off-policy data. This design effectively avoids the risk of disrupting established model patterns and overfitting to expert data in traditional SFT-RL integration.

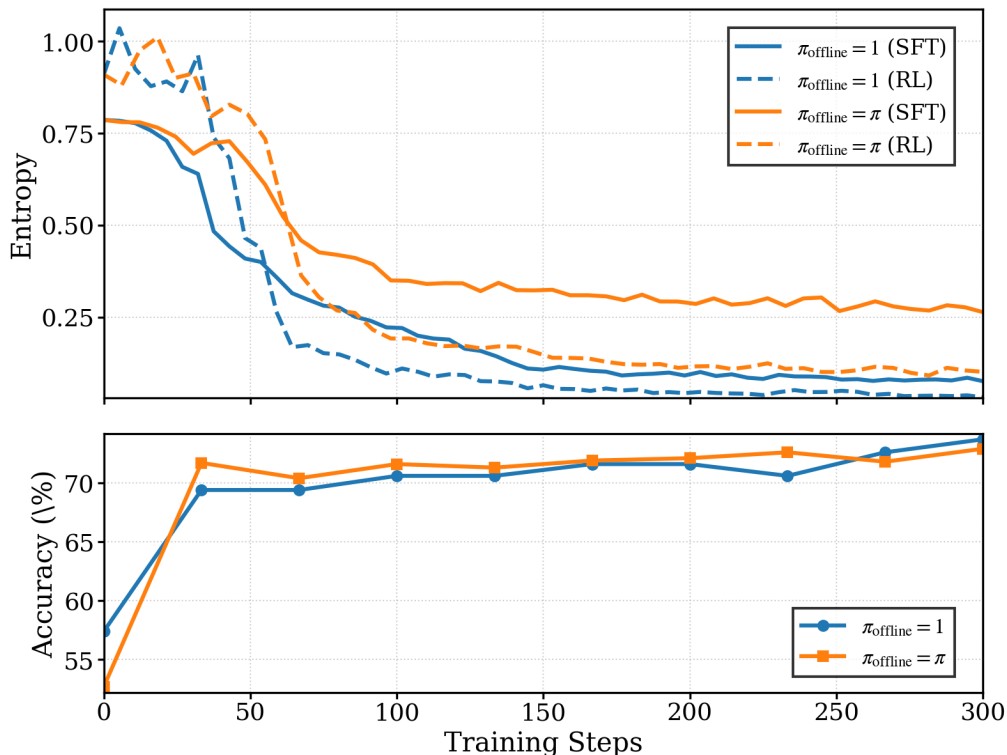

Figure 10: SFT/RL entropy and accuracy over training steps under two choices of $\pi^{\text{offline}}$: (i) $\pi^{\text{offline}} = 1$ and (ii) $\pi^{\text{offline}} = \pi$. The entropy collapse and accuracy drop illustrate the sensitivity of training dynamics to the offline policy specification.

GTA (Zeng et al., 2025) proposes a three-stage supervised-guided RL framework (Guess-Think-Answer) for text classification with LLMs. In the Guess stage, the model generates a provisional prediction optimized via cross-entropy loss (leveraging SFT's training efficiency); the Think stage refines this guess to improve decision logic; and the Answer stage produces the final output, with RL rewards shaping both the final result and the overall GTA structure. To address gradient conflicts between SFT and RL signals, it further employs loss masking and gradient constraints, achieving faster convergence than pure RL and a higher performance ceiling than pure SFT.

For the underlying mechanism of SFT-RL synergy, Jin et al. (Jin et al., 2025) challenge the over-simplified claim "SFT memorizes, RL generalizes" by conducting full-parameter fine-tuning experiments on LLaMA-3.2-11B and Qwen-2.5-7B. They discover that SFT suffers from *OOD forgetting*—OOD)performance peaks in the early stage of SFT and declines thereafter—and that RL only recovers the lost OOD reasoning capability (rather than creating fundamentally new OOD ability), with recovery effective only within a specific range of SFT checkpoints. Through Singular Value Decomposition (SVD) analysis, they further identify that the rotation of singular vectors (instead of stable singular values) is the core factor driving OOD forgetting (from SFT's hard alignment of parameter directions) and recovery (from RL's soft re-alignment).

Beyond the decoupled two-stage SFT-RL paradigm, Chen et al. (Chen et al., 2025d) proposes a co-operative training framework based on bilevel optimization. The lower level performs RL policy updates while receiving real-time SFT supervision (ensuring RL exploration does not deviate from task objectives), and the upper level explicitly maximizes "cooperative gain"—the performance advantage of joint SFT-RL training over standalone RL. This design strengthens the interaction between SFT and RL, addressing RL's inefficiency due to trial-and-error while enhancing overall reasoning performance.

At the theoretical level, Lv et al. (Lv et al., 2025) establishes a unified view of LLM post-training by deriving a *Unified Policy Gradient Estimator*. It demonstrates that SFT, RL, and other mainstream

post-training methods are all instantiations of the gradient of a common objective, differing only in data distribution assumptions and bias-variance tradeoffs; the estimator consists of four inter-changeable components (stabilization mask, reference policy denominator, advantage estimate, and likelihood gradient). Motivated by this theory, they propose Hybrid Post-Training (HPT), which dynamically selects offline demonstration signals (for SFT) and online exploration signals (for RL) to balance effective exploitation of expert data and stable exploration without sacrificing learned reasoning patterns, achieving consistent improvements across six mathematical reasoning benchmarks and two OOD suites.

## J OOD BENCHMARKS

**ARC-C (AI2 Reasoning Challenge - Challenge)** A dataset of grade-school level science questions that require commonsense reasoning and knowledge application. The dataset consists of multiple-choice questions designed to be easy for humans but challenging for AI systems, testing the model's ability to apply scientific knowledge and reasoning in everyday contexts beyond pure mathematical domains.

**GPQA-D (Graduate-Level Google-Proof Q&A - Diamond)** A challenging benchmark designed to evaluate advanced reasoning capabilities in scientific domains. GPQA consists of 448 multiple-choice questions, with the "Diamond" subset containing 198 hard problems in biology, physics, and chemistry. These "Google-proof" questions require graduate-level domain expertise and are specifically designed to be difficult to answer even with internet search.

**MMLU-Pro (Massive Multitask Language Understanding - Professional)** An enhanced and more robust version of the original MMLU benchmark. MMLU-Pro covers 57 subjects across disciplines such as mathematics, history, law, and medicine. It not only assesses factual knowledge but also the model's capacity to apply this knowledge in context-specific scenarios. Compared to MMLU, MMLU-Pro increases question difficulty and reduces potential shortcuts while maintaining broad academic coverage.

