# OpenReview forum: "Recall-Extend Dynamics: Enhancing Small Language Models through Controlled Exploration and Refined Offline Integration"
_ICLR.cc/2026/Conference — ICLR 2026 Conference Withdrawn Submission_

### Official Review · Reviewer_1zfK · 2025-10-28

**Soundness:** 2
**Presentation:** 2
**Contribution:** 2
**Rating:** 2
**Confidence:** 3

**Summary:**

This submission presents Recall-Extend Dynamics (RED), a training paradigm designed to enhance the reasoning abilities of small language models (SLMs) by effectively combining reinforcement learning with verifiable rewards (RLVR) and supervised fine-tuning (SFT). The authors conceptualize RLVR as a “Recall” mechanism that optimizes reasoning paths within the model’s existing knowledge space, while they conceptualize SFT as acting as an “Extend” mechanism that broadens exploration by distilling reasoning patterns from larger teacher models.

RED dynamically regulates the interaction between these phases through entropy-based exploration control, measuring changes in model entropy to adjust the relative influence of RL and SFT during training. When exploration entropy is small, SFT’s contribution is amplified to inject new reasoning diversity. In the opposite case, RL is weighted more heavily to refine learned reasoning strategies. This approach avoids the static or staged scheduling used in prior methods and provides a continuous, self-regulating balance between exploration and exploitation.

The authors also introduce an accuracy-aware policy shift mechanism that adaptively adjusts how the model integrates offline distillation data. Specifically, the authors estimate the off-policy probability of the current trajectory for a larger teacher model based on sample correctness: high-accuracy samples are used to reinforce the model’s own policy, while low-accuracy ones encourage imitation of the teacher’s policy. This approach yields theoretical guarantees of gradient stability, unbiased learning, and asymptotic consistency while mitigating entropy collapse.

Finally, the authors also run experiments on mathematical reasoning benchmarks (MATH500, AIME24/25, AMC, Minerva, and Olympiad), and compare their performance with that of prior SFT, RL, and hybrid frameworks.

**Strengths:**

The problem of reasoning in small language models is important and well-motivated. Moreover, the authors propose a practical framework which comes with some nice built-in properties.

**Weaknesses:**

With that being said, the authors’ contribution is somewhat incremental when compared to previous work in this area. Additionally, the empirical results are also somewhat incremental/inconclusive as to whether (and by how much) this method improves over baselines.

The theoretical results could also be proven more rigorously. For example, the “proof” of Proposition B.3 reads more like a proof sketch than an actual proof. Finally, the paper was hard to follow at times (e.g. many long equations are presented without much intuition).

**Questions:**

Could you provide a more formal proof of Proposition B.3?

---

### Official Review · Reviewer_ZeYh · 2025-10-30

**Soundness:** 2
**Presentation:** 1
**Contribution:** 2
**Rating:** 4
**Confidence:** 4

**Summary:**

This paper presents Recall-Extend Dynamics (RED), a framework designed to enhance the reasoning capabilities of small language models (SLMs) by unifying supervised fine-tuning (SFT) and reinforcement learning with verifiable rewards (RLVR). The core idea is to balance Recall (refining reasoning within the model’s existing competence through RL) and Extend (expanding reasoning capacity through SFT). This balance is achieved via Dynamic Entropy Regulation and an Accuracy-Aware Policy Shift Mechanism.

**Strengths:**

1. The proposed Recall-Extend dichotomy is conceptually elegant and provides an intuitive lens for understanding SFT–RL synergy in small models.

2. The proposed entropy-based regulation and accuracy-aware policy offset are both theoretically analyzed with proofs of stability.

3. RED achieves consistent gains in reasoning accuracy and efficiency across multiple datasets.

**Weaknesses:**

1. The paper’s core contributions, Dynamic Entropy Regulation and Accuracy-Aware Policy Shift, represent well-motivated but incremental refinements to existing unified SFT–RL frameworks, rather than a fundamentally new paradigm.

2. Figure 1, and Figure 5 report accuracy or entropy trends, but the paper does not specify on which dataset or benchmark these results were obtained.

3. The paper claims that RED improves efficiency but provides no quantitative comparison (e.g., GPU hours or rollout time) with other methods.

4. Experiments are limited to mathematical reasoning; validation on general or multimodal reasoning tasks would strengthen the empirical evidence.

**Questions:**

1-4. See weaknesses above.

5. Have the authors evaluated RED beyond mathematical and scientific reasoning, for example, on commonsense or multimodal datasets?

---

### Official Review · Reviewer_B8QP · 2025-11-01

**Soundness:** 2
**Presentation:** 2
**Contribution:** 1
**Rating:** 2
**Confidence:** 5

**Summary:**

This paper provides a new workflow in RLVR called Recall-Extend Dynamics (RED), which combines the GRPO loss and the SFT loss with modifications. An advantage term and an importance-sampling term are added into the SFT loss. This paper proposes a novel way to design the importance-sampling term. The experimental results are evaluated on several benchmarks, though only on one base model.

**Strengths:**

- The design of $\pi^{offline}$ is novel to the reviewer and the figure 6 looks good.
- The results are evaluated on several benchmarks, though only on one base model.

**Weaknesses:**

The general algorithm is not novel, which is a combination of GRPO objective and SFT objective with minimal modifications. However, the intuition of the modifications is not well explained. Please see the "Questions" section.

The main experiments are only conducted on Qwen2.5-math-1.5B model, making the empirical improvement less convincing.

The theoretical part is highly informal, and the reasons are listed as follows:
1. The proof of "Entropy Preservation" property in Proposition 1 is not rigorous. Please formally state the implicit assumptions and provide  rigorous proof. Otherwise it's not a proposition. For example, 1) the first-order approximation of $\Delta H(\pi_\theta)$ is wrong: $\Delta H(\pi_\theta)$ is a scalar while the formula given by the authors is a vector, and the expectation over states is missed; 2) $\nabla_\theta \pi_\theta(a'\vert s_t)$ is a vector, and the authors cannot claim that vector to be negative; 3) the authors need to assume the existence of a action $a$, s.t. $\log\pi_\theta(a)\rightarrow -\infty$, which is non-trivial;
2. Eq.9 is trivial when $\beta=0$. While for $\beta\rightarrow 0$ (which is covered in Proposition 2), to use Dominated convergence theorem, you need assumption on the boundness of $\nabla_\theta\log \pi_\theta(a_t|s_t)$. Please state it formally.
3. The proof of Proposition 3 is informal and not self-contained.

Additionally, the theoretical results can only provide weak results, such that the "policy" will converge to optimal policy a.s., which can be guaranteed by most RL algorithms. More important properties such as convergence rates and sample complexity are not covered.

Therefore, the theoretical part is of minimal contribution and should be further revised.

As a conclusion, the reviewer thinks that this paper cannot be accepted to a top-tier venue like ICLR.

**Questions:**

- Why should there be a length normalization term for the SFT component? The Eq 4 is not maximizing $\pi(o_{sft}\vert q)$ but maximizing $(\pi(o_{sft}\vert q))^{1/|o_{sft}|}$.
 - Why "By adding an advantage term, RL with SFT loss can be easily transformed into on-policy format"? What's the intuition of this choice? And what's $\hat A_{i,t}$ in the second line of Eq 7? There seems to be no $i$ in the SFT component.
- What's the definition of $r_{group}$?

---

### Official Review · Reviewer_6Qd9 · 2025-11-01

**Soundness:** 1
**Presentation:** 2
**Contribution:** 1
**Rating:** 2
**Confidence:** 4

**Summary:**

The paper targets reasoning improvements for small language models by concurrently using offline distillation (SFT) and online RL with verifiable rewards. The model highlights two designs that balance different objectives. Dynamic entropy regularization balances SFT and RL objectives. An accuracy-aware mechanism balances sample weights for distilled trajectories and the model’s own collected trajectories.

**Strengths:**

1. The idea of using entropy to balance different objectives is intuitive and easy to implement.

2. The model is compared with competitive baselines, including unified and stage-wise frameworks.

**Weaknesses:**

1. Optimizing SFT and RL objectives concurrently can induce conflicting gradient signals and non-stationary targets, especially when the SFT teacher distribution and the RL policy are far from each other. This can exacerbate training instability that RL-based approaches face.

2. The method is framed as targeted at small language models, but the design and analysis appear model-size agnostic. The paper does not isolate challenges unique to SLM fine-tuning (e.g., weaker in-context learning, shorter context windows, tighter compute/memory budgets, instability under long CoT) nor demonstrate behaviors that fundamentally differ from larger models.

3. Lines 195-196 directly remove the KL divergence in GRPO and make the assumption that the ratio of current and reference policy is always within clipping range. This assumption needs to be further justified.

**Questions:**

Please see weaknesses.

Further questions:

1. What is $ r_{group}$ in equation 8 and how is it related to $\beta$? Is it the high accuracy or high error rate indicator of a group of samples? How is it obtained in practice?

2. What is proposition 1 (Stability and Adaptivity); 2. Robustness trying to prove? Why robustness can be inferred given the current policy has a zero probability of sampling action a given state s.

---

### Note · Authors · 2025-11-12

I have read and agree with the venue's withdrawal policy on behalf of myself and my co-authors.